# Using observation-level random effects to model overdispersion in count data in ecology and evolution

Xavier A. Harrison

Institute of Zoology, Zoological Society of London, London, UK

## ABSTRACT

Overdispersion is common in models of count data in ecology and evolutionary biology, and can occur due to missing covariates, non-independent (aggregated) data, or an excess frequency of zeroes (zero-inflation). Accounting for overdispersion in such models is vital, as failing to do so can lead to biased parameter estimates, and false conclusions regarding hypotheses of interest. Observation-level random effects (OLRE), where each data point receives a unique level of a random effect that models the extra-Poisson variation present in the data, are commonly employed to cope with overdispersion in count data. However studies investigating the efficacy of observation-level random effects as a means to deal with overdispersion are scarce. Here I use simulations to show that in cases where overdispersion is caused by random extra-Poisson noise, or aggregation in the count data, observation-level random effects yield more accurate parameter estimates compared to when overdispersion is simply ignored. Conversely, OLRE fail to reduce bias in zero-inflated data, and in some cases increase bias at high levels of overdispersion. There was a positive relationship between the magnitude of overdispersion and the degree of bias in parameter estimates. Critically, the simulations reveal that failing to account for overdispersion in mixed models can erroneously inflate measures of explained variance ($r^2$), which may lead to researchers overestimating the predictive power of variables of interest. This work suggests use of observation-level random effects provides a simple and robust means to account for overdispersion in count data, but also that their ability to minimise bias is not uniform across all types of overdispersion and must be applied judiciously.

> "Overdispersion is the polite statistician's version of Murphy's Law: if something can go wrong, it will"
> *Crawley* (*2007*, p. 522)

## INTRODUCTION

Count data are extremely common in the fields of evolutionary biology and ecology; researchers are often interested in quantifying the factors affecting variables such as how many offspring an individual produces, counts of parasite load, abundance of

Corresponding author
Xavier A. Harrison,
xav.harrison@gmail.com

species within and between habitats, or the frequency with which individuals perform certain behaviours. Perhaps the most common method employed to model count data is to assume the data approximate a Poisson distribution, and specify statistical models accordingly (e.g., *Bolker et al., 2009*). However, a persistent problem with Poisson models is that they often exhibit overdispersion, where the variance of the response variable is greater than the mean (*Hilbe, 2011*), resulting in a poor fit to the data. Accounting for overdispersion when it is present is critical; failing to do so can lead to biased parameter estimates (*Crawley, 2007*; *Hilbe, 2011*), and cause researchers to erroneously conclude that variables have a meaningful effect when in fact they do not (*Crawley, 2007*; *Richards, 2008*; *Zuur et al., 2009*). The necessity for accurate biological inference therefore demands that we employ tools to both identify and adequately deal with overdispersion to minimize the risk of Type I error (*Hilbe, 2011*). One manner in which overdispersion is dealt with involves the use of observation-level random effects (OLRE), which model the extra-Poisson variation in the response variable using a random effect with a unique level for every data point. However, data on the efficacy of OLRE as a tool to accurately model overdispersion and recover correct parameter estimates are relatively scarce (but see *Bolker et al., 2009*; *Kery, 2010*). This paper uses a simulation approach to address the shortfall in our understanding of the ability of OLRE to cope with the types of overdispersion commonly encountered in mixed models of count data in ecology and evolution.

Overdispersion occurs primarily for two reasons; 'apparent overdispersion' (*Hilbe, 2011*) arises when models have been poorly specified, for example by failing to include important predictors, interactions between predictors that have already been measured, or by specifying the incorrect link function (*Hilbe, 2011*). Conversely, 'real overdispersion' can arise when there is clustering in the count data, meaning observations are not truly independent of one another (*Hilbe, 2011*), when there is an excess number of zeroes in the data (zero-inflation) (*Zuur et al., 2009*), or when the variance of the response truly is greater than the mean (i.e., is not accurately described by a Poisson process). In cases of both real and apparent overdispersion, the fit of the model to the data will be poor, even if the model uncovers significant results. It is worth noting that these causes are not mutually exclusive, and in fact models of interest may suffer from several of these issues simultaneously. This paper will deal specifically with three cases of real overdispersion: data that contain extra-Poisson noise, data where the counts are non-independent (aggregated), and zero-inflated data. There are several modeling tools available for dealing with overdispersion, including negative binomial models, also known as Poisson-gamma mixture models, to deal with aggregation in count values (e.g., *Hilbe, 2011*) and zero-inflated Poisson (ZIP) models for an excess of zeroes in the data (*Zuur et al., 2009*; *Zuur, Saveliev & Ieno, 2012*). This paper will deal exclusively with Poisson models containing OLRE (also known as Poisson-lognormal models; *Elston et al., 2001*) as a tool for modeling overdispersion.

## Quantifying overdispersion in mixed models

In order to satisfy the assumption of Poisson errors, the residual deviance of a candidate model should be roughly equal to the residual degrees of freedom (e.g., *Crawley, 2007*,

p. 540). The ratio of these two values is referred to as the dispersion parameter, and values >1 indicate overdispersion. Calculating this value is straightforward in a Generalized Linear Model (GLM) context (i.e., models without random effects), and many software packages such as R (*R Core Team, 2014*) will calculate this value automatically for GLMs. For Generalized Linear Mixed Models (GLMMs), the situation becomes more complex due to uncertainty in how to calculate the residual degrees of freedom (d.f.) for a model that contains random effects. A single random effect is estimated to use between 1 (by estimating the standard deviation of the distribution) and $N - 1$ d.f., where $N$ is the number of levels of the random effect (i.e., by estimating the mean value for each level). More sophisticated approximations of the residual d.f. are also available (see *Bolker et al., 2009*, Box 3), but not discussed here.

For mixed models, the dispersion parameter can be calculated as the ratio of the sum of the squared Pearson residuals to the residual degrees of freedom (e.g., *Zuur et al., 2009* p. 224). Moreover, the sum of squared Pearson residuals should approximate a Chi-squared distribution with n-p degrees of freedom (*Bolker et al., 2009*), and so we can also test for the 'significance' of overdispersion in models. *Bolker et al. (2009)* provide a useful R function for calculating the dispersion parameter using the squared Pearson residuals (assuming 1 d.f. lost for every random effect), and providing a *p* value for the test. However, there are two caveats for this test: (1) the uncertainty over how to calculate residual degrees of freedom (the denominator in the ratio); and (2) the fact that models with small sample sizes will yield sum of squares (SS) that may not approximate a Chi-squared distribution. Both of these issues can be avoided if one calculates the overdispersion parameter by parametric bootstrapping. Each replicate of a parametric bootstrap generates a new dataset (response variable) based on the parameter estimates and distributional assumptions (i.e., Poisson) in the model of interest. If the model is unbiased, the new response data generated under parametric bootstraps should be comparable to the 'real' data collected by a researcher. If the model was a poor fit, the magnitude of the residuals of the original model should be much larger than the residuals generated from parametric bootstrapping. The ratio of the residual SS of the original model to the mean of the bootstrapped residual SS is equivalent to the dispersion parameter. The advantage of the parametric bootstrap approach, aside from not relying on calculating residual d.f., is that it can be used to generate confidence intervals for the magnitude of the dispersion parameter. R code demonstrating the use of parametric bootstrapping to calculate overdispersion is supplied as Supplemental Information 1.

## Aims

Overdispersion is a serious problem because it can bias both the means and standard errors of parameter estimates (*Hilbe, 2011*). Unfortunately, few studies reporting significant predictors assess the degree of overdispersion in their models (*Richards, 2008*), despite explicit guidance on how to calculate dispersion parameters for GLMs (see *Crawley, 2007*; *Zuur et al., 2009*) and GLMMs (see *Bolker et al., 2009*). Moreover, recently *Nakagawa & Schielzeth (2013)* developed a method to calculate a measure of explained variance ($r^2$)

for GLMMs, but it is not clear how overdispersion in models affects estimates of $r^2$ when overdispersion is known to bias parameter estimates away from their true values (*Hilbe, 2011*; *Kery, 2010*). Theoretically, biased parameter estimates caused by overdispersion could also yield biased estimates of proportion of explained variance, and this is a possibility that warrants further investigation.

This paper will test the robustness of the use of OLRE to model real overdispersion in biological data. The primary aims of this paper are to:

(i) Test the ability of OLRE to recover precise parameter estimates from data generated under scenarios where overdispersion is caused by: (a) the variance of the response being greater than the mean due to extra-Poisson 'noise', (b) an excess of zeroes in the response (zero-inflation), or (c) aggregation (non-independence) of counts among samples.

(ii) Assess the influence of overdispersion on estimates of explained variance ($r^2$) in models where overdispersion is either ignored, or explicitly dealt with using OLRE.

## MATERIALS AND METHODS

All simulations in this paper were conducted in R v3.1.0 (*R Core Team, 2014*) using the *lme4* package (v1.1-6; *Bates et al., 2014*) to both simulate data, and fit GLMMs to those data. R code for all simulations, models and graphs are provided in Supplemental Information 1, including some convenience functions to handle the output of the simulations. I also point the reader to the supplementary online material for *Bolker et al. (2009)* which contains several convenience functions for mixed models, including one to calculate the point estimate of overdispersion in GLMMs with corresponding p values.

This paper deals with overdispersion arising from three principle sources: extra-Poisson 'noise' in the data ('Noise' Simulations'), zero-inflation ('Zero Inflation' Simulations) and aggregation in the counts ('Negative Binomial' Simulations). For all sources, I consider a hypothetical species where a researcher has counted the number of parasites on 50 individuals from each of 10 populations. General details of the simulation methodology are below, whilst values employed for each of the parameters in the simulations are listed separately.

### Extra-Poisson noise simulations

Following *Zuur et al. (2009)*, the Poisson distribution function is given by

$$f(y; \mu) = \frac{\mu^y \times e^{-\mu}}{y!}. \tag{1}$$

This formula specifies the probability of observing a single value $y$ given a mean of $\mu$, where $y \geq 0$ and $y$ is an integer. For a set of observations $Y$ arising from a Poisson distribution, the mean and variance are identical:

$$E(Y) = \mu \quad \text{and} \quad var(Y) = \mu. \tag{2}$$

Overdispersion can arise because extra variation in the data destabilizes the mean–variance relationship i.e., $E(Y) = var(Y) = \mu$ in Eq. (2). In a standard Poisson model, we can place a

model on the mean $\mu$ as follows:

$$\mu_i = e^{\alpha + \beta x_i} \tag{3}$$

where $\alpha$ is the intercept, $\beta$ is the slope term, and $x_i$ is the covariate value for observation $i$ for a hypothetical predictor (e.g., body size). However, often the value of $\mu$ can be a function of additional, unmodelled sources of variation as follows:

$$\mu_i = e^{\alpha + \beta x_i + \varepsilon_i} \tag{4}$$
$$\varepsilon_i \sim N(0, \sigma_\varepsilon) \tag{5}$$

where $\varepsilon_i$ represents additional variation in the linear predictor specific to each observation, drawn from a normal distribution with mean 0 and SD $= \sigma_\varepsilon$. Because researchers often do not consider the additional sources of variation in the data such as $\varepsilon_i$, i.e., they model the mean using Eq. (3), not Eq. (4), the variance of the data will be much larger than predicted by the model. To explore the effect of overdispersion caused by extra-Poisson noise (Eq. (4)), I simulated data where the number of parasites on an individual host increases as a function of body size, but added random noise to the linear predictor so that the resulting data were not well approximated by a Poisson distribution. The following example is slightly more complex than Eq. (4), as I assume that mean parasite load varies among populations (i.e., introduce a random effect of population). I assumed a homogenous slope of the effect of body size across the populations.

$$C_{ij} \sim Pois(\mu_{ij}) \tag{6}$$
$$\mu_{ij} = e^{\alpha_{ij} + \beta x_i + \varepsilon_i} \tag{7}$$
$$\alpha_j \sim N(\mu_\alpha, \sigma_\alpha) \tag{8}$$

where $C_{ij}$ is the count value for individual $i$ in population $j$, $\mu_{ij}$ is the mean count for individual $i$ in population $j$, $\beta$ is the slope parameter for the effect of body size, $x_i$ is the body size measurement of individual $i$. Population-specific intercepts ($\alpha_j$) are drawn from a normal distribution with mean $\mu_\alpha$ and standard deviation $\sigma_\alpha$. Values for $\varepsilon_i$ follow Eq. (5) above. Values were set at $\mu_\alpha = -0.5$, $\sigma_\alpha = 0.5$ and $\beta = 0.1$ for all models. $\sigma_\varepsilon$, in this example is the 'noise parameter': the larger the value for this parameter, the greater the noise:signal ratio in the count data, and the greater the degree of overdispersion that will be present. I simulated 10 scenarios where $\sigma_\varepsilon$ ranged from 0.1 to 1 in increments of 0.1, and thus the variance ratio of the noise parameter SD: random intercept SD varied from 0.2 to 2 (0.1/0.5 to 1/0.5).

## Zero-inflated Poisson simulations

For the Zero Inflation simulations, I simulated count data as above, assuming an effect of body size on number of parasites. However I also introduced stochastic variation in parasite load, where some individuals in the study have a parasite count of 0 irrespective of their body size. An excess of zeroes in the dataset can arise for two reasons: (i) individuals may be resistant to a particular parasite because of immunogenetic variability that protects

them (e.g., *Owen, Nelson & Clayton, 2010*); or (ii) some individuals may indeed be parasitised, but the researcher may fail to see any parasites on the hosts (i.e., they are false zeroes). I do not distinguish between these two cases here, but see *Martin et al. (2005)*; *Zuur et al. (2009)* and *Zuur, Saveliev & Ieno (2012)* for an in depth discussion on the various methods of modelling both true and false zeroes. Following *Zuur et al. (2009)*, the zero-inflated Poisson (ZIP) probability function is given by:

$$f\left(y_i = 0\right) = \pi_i + (1 - \pi_i) \times e^{-\mu_i} \tag{9}$$

$$f\left(y_i | y_i > 0\right) = \pi_i + (1 - \pi_i) \times \frac{\mu^{y_i} \times e^{-\mu_i}}{y_i!}. \tag{10}$$

Here, the probability of measuring additional zeroes than would be predicted by an ordinary Poisson distribution is controlled by $\pi_i$, where higher values of $\pi_i$ yield a greater frequency of zeroes. Note that parasite resistance (zero counts) can still arise when $\pi = 1$, from the Poisson portion of the model. As for above, all models used $\mu_\alpha = -0.5, \sigma_\alpha = 0.5$ and $\beta = 0.1$. I simulated 10 scenarios where $\pi$ ranged from 0 to 0.5 in increments of 0.05. A value of $\pi = 0$ was used as a control to check that the model collapsed to an ordinary Poisson model (i.e., no excess zeroes) and recovered correct parameter estimates. Values for $\pi$ higher than 0.5 (>50% chance of measuring a zero) caused problems with model convergence and so were not investigated. In this model, the zero-inflation occurs at random across the whole dataset with a fixed probability ($\pi$). For more complex and elegant examples of zero-inflated data, I recommend *Kery (2010)*, which provides cases where the zero-inflation occurs systematically across the data and can be thus be modelled with covariates.

## Negative binomial simulations

The negative binomial distribution is derived from what is referred to as a Poisson-gamma mixture distribution (*Hilbe, 2011*) where one assumes that the data $Y$ are Poisson distributed, but that the mean $\mu$ comes from a Gamma distribution (*Zuur et al., 2009*). The Poisson distribution assumes that the counts $Y$ are independent from one another (*Hilbe, 2011*), but when this assumption is violated and there is, for example, aggregation in the counts, the Poisson-gamma mixture model can account for this non-independence (*Hilbe, 2011*). Following *Zuur et al. (2009)*, the negative binomial distribution can be parameterized as:

$$f\left(y; k, \mu\right) = \frac{\Gamma(y + k)}{\Gamma(k) \times \Gamma(y + 1)} \times \left(\frac{k}{\mu + k}\right)^k \times \left(1 - \frac{k}{\mu + k}\right)^y \tag{11}$$

where the mean and variance of $Y$ are:

$$E(Y) = \mu \quad \text{and} \quad var(Y) = \mu + \frac{\mu^2}{k}. \tag{12}$$

Thus, $k$ controls the level of overdispersion in the dataset. As $k$ becomes large relative to $\mu$, overdispersion is minimized and the model collapses to an ordinary Poisson distribution

(*Zuur et al., 2009*). Conversely smaller values of $k$ yield increasingly greater levels of overdispersion. For the Negative Binomial simulations, I simulated count data from a negative binomial distribution using the function 'rnegbin' in the MASS package (*Venables & Ripley, 2002*) in R, using values of $k$ from 5 to 0.25 in steps of roughly 0.5.

For all three scenarios (noise, zero-inflation and negative binomial), the body size variable was drawn randomly from a normal distribution with mean 30 and SD 4 for every iteration of the simulations.

## Quantifying overdispersion in mixed models

To verify that the simulations employed resulted in overdispersed count data. For each of the 10 Noise and Zero-Inflation datasets, I fitted a GLMM to the data, specifying a random intercept for population, body size as a predictor, Poisson errors and a log link. I then performed 100 parametric bootstraps of the model using the 'bootMer' function in *lme4*, specifying a function to calculate the sum of squared Pearson residuals for each of the 100 bootstrapped models. I then calculated the ratio of the original model's SS residuals to each of the 100 bootstrapped SS residuals, and calculated a mean and 95% confidence interval for these values. This ratio is equivalent to the dispersion parameter, but does not require calculation of the residual d.f. of the model. All parametric bootstrapped dispersion parameter estimates were in broad agreement with the point estimate calculated using the sum of the squared Pearson residuals. These data are shown in Supplemental Information 1.

## Overdispersion and biased parameter estimates

For each value of $\pi$ in the Zero-Inflated data, each value of $k$ in the Negative Binomial Simulations, and each value of $\sigma_\varepsilon$ in the Noise data, I simulated 100 replicated datasets. For each iteration, I fitted both a 'naive' GLMM containing only a random intercept for population and body size as a predictor, in addition to a second GLMM identical to the first but also including an observation-level random effect. Using R code notation, these are of the general form:

$$Count \sim BodySize + \left(1|Population\right) \qquad \text{'Naïve' Model}$$
$$Count \sim BodySize + \left(1|Population\right) + (1|OLRE) \qquad \text{'OLRE Model'}$$

where *OLRE* refers to the presence of an observation-level random effect in addition to the *Population* random effect. Hereafter these two models will be referred to as 'Naive' and 'OLRE' models, respectively. Both models had Poisson errors and a log link. The OLRE variable was a sequence of numbers from 1 to 500, acting as 'observation ID' for each row in the dataset. For both models at each iteration I stored the estimates of $\mu_\alpha$, $\sigma_\alpha$, and $\beta$, in addition to $\sigma_\varepsilon$ for the Noise simulations, $\pi$ for the zero-inflated simulations, and $k$ for the Negative Binomial simulations. These 100 estimates of each parameter were used to plot means and 95% confidence intervals. For the Zero-Inflation and Negative Binomial models, I verified recovery of the correct parameter estimates with mixed models fitted using the *glmmADMB* package (v0.8.0; *Skaug et al., 2014*; *Fournier et al., 2012*) in R using zero-inflated Poisson and negative binomial error distributions, respectively.

## Overdispersion and $r^2$ calculations

In addition to storing the parameter estimates as above, for each iteration I also calculated the $r^2$ values for the Naive and the OLRE models using the methods detailed in *Nakagawa & Schielzeth (2013)*. Following *Nakagawa & Schielzeth (2013)*, I calculated the marginal $r^2$ (variance explained only by fixed effects) as:

$$\frac{\sigma^2_{\text{fixed}}}{\sigma^2_{\text{fixed}} + \sigma^2_{\text{random}} + \sigma^2_{\text{residual}}} \tag{13}$$

and the conditional $r^2$ (variance explained by both fixed and random effects) as:

$$\frac{\sigma^2_{\text{fixed}} + \sigma^2_{\text{random}}}{\sigma^2_{\text{fixed}} + \sigma^2_{\text{random}} + \sigma^2_{\text{residual}}} \tag{14}$$

where $\sigma^2_{\text{fixed}}$ is the fixed effects variance of the model, $\sigma^2_{\text{random}}$ is the random effects variance of the model (or sum of random effects variances for a model with multiple random intercept terms), and $\sigma^2_{\text{residual}}$ is the residual variance. Specific details are found in *Nakagawa & Schielzeth* (*2013*, equations 29 and 30) and the accompanying R code for the paper. In the simulations I calculated and stored $\sigma^2_{\text{fixed}}, \sigma^2_{\text{random}}$, and $\sigma^2_{\text{residual}}$ separately to investigate the effects of overdispersion on each of these components of the $r^2$.

## RESULTS

### Noise simulations

At low levels of overdispersion (small vales of $\sigma_\varepsilon$), both the Naive model and the OLRE model recovered near identical estimates for all parameters. However, as overdispersion increased, the degree of bias in $\mu_\alpha$, the mean of the random intercept for population ID, increased substantially in the Naive models (Fig. 1A). Accurate estimates of the slope parameter were recovered from both the Naive and OLRE models, irrespective of the magnitude of $\sigma_\varepsilon$ (Fig. 1B), but the standard errors for all estimates were smaller by up to a factor of 5 in the Naive models (Table 1A). Similarly, both models estimated sensible values of the SD of the random intercept at all values of $\sigma_\varepsilon$ ($\sigma_\alpha \sim 0.45$), although the true value was slightly higher (0.5). There was some indication in the data that as the magnitude of $\sigma_\varepsilon$ increased, the estimates of $\sigma_\alpha$ started to diverge, with values from the Naive model being upwardly inflated (Fig. 1C).

### Zero-inflation simulations

At low levels of overdispersion ($\pi = 0$–0.2), both the Naive model and the OLRE model produced identical parameter estimates for $\mu_\alpha, \beta$ and $\sigma_\alpha$ (Fig. 2). However, further increases in $\pi$ lead to downward bias in the values for all three parameters. The degree of this bias was *greater* for the OLRE models than for the Naive models, especially for $\sigma_\alpha$ (Fig. 3C; $\sigma_\alpha = 0.05$ at $\pi = 0.5$, true value of $\sigma_\alpha = 0.5$). The large reductions observed for $\sigma_\alpha$ were mirrored by large increases in the magnitude of $\sigma_\varepsilon$. With respect to $\sigma_\alpha$ and $\beta$, the Naive model recovered the correct parameter estimates irrespective of the degree
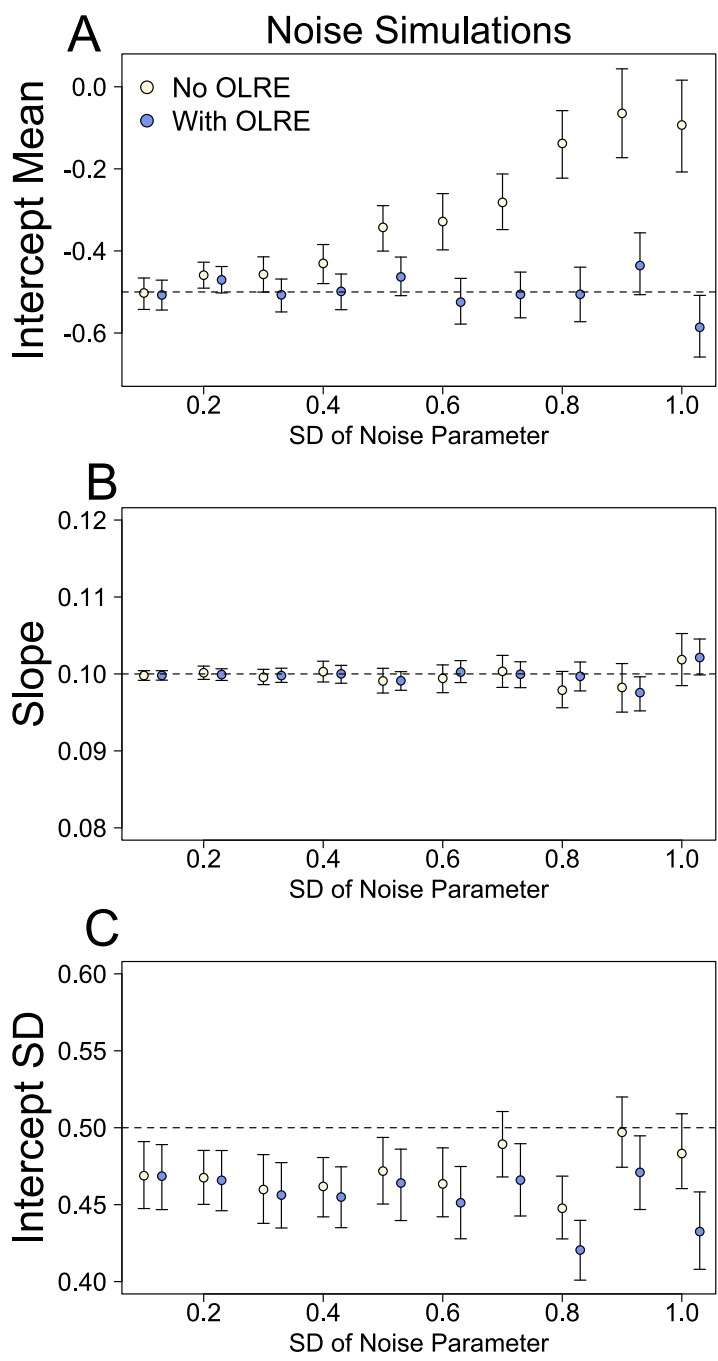

**Figure 1 Model parameters for the intercept mean (A), slope of the effect of body size (B) and intercept standard deviation (C) generated under various levels of overdispersion for the noise simulations.** Light circles represent the mean values of the Naive models, where overdispersion was ignored. Blue circles represent the models containing observation-level random effects. Error bars are 95% confidence intervals of the mean as estimated by bootstrapping. Dashed horizontal lines denote the true (simulated) parameter values.

**Table 1 Estimates of the standard errors of the slope and intercept parameters estimated under three scenarios of overdispersion (zero inflation, extra-poisson noise, and negative binomial).** Vales of $\pi$, $\sigma_\varepsilon$ and $k$ for the three scenarios, respectively, correspond to increasing levels of overdispersion moving down the table. The 'Ratio' column indicates the ratio of the parameter estimate where overdispersion was modelled with observation-level random effects (OD) to the estimate where overdispersion was ignored (Naive).

| | Naïve slope | OD slope | Ratio | Naïve intercept | OD intercept | Ratio |
|---|---|---|---|---|---|---|
| | | | **(A) Zero inflation** | | | |
| $\pi$ | | | | | | |
| 0 | 0.003 | 0.003 | **1.02** | 0.180 | 0.181 | **1.00** |
| 0.05 | 0.003 | 0.005 | **1.60** | 0.179 | 0.213 | **1.19** |
| 0.11 | 0.003 | 0.007 | **2.32** | 0.179 | 0.267 | **1.49** |
| 0.17 | 0.003 | 0.010 | **3.03** | 0.179 | 0.334 | **1.86** |
| 0.22 | 0.003 | 0.013 | **3.89** | 0.183 | 0.424 | **2.31** |
| 0.27 | 0.004 | 0.014 | **4.10** | 0.187 | 0.450 | **2.41** |
| 0.33 | 0.004 | 0.017 | **4.75** | 0.188 | 0.532 | **2.82** |
| 0.38 | 0.004 | 0.022 | **5.90** | 0.197 | 0.686 | **3.48** |
| 0.4 | 0.004 | 0.028 | **7.07** | 0.193 | 0.869 | **4.50** |
| 0.5 | 0.004 | 0.031 | **7.42** | 0.204 | 0.946 | **4.63** |
| | | | **(B) Extra-Poisson noise** | | | |
| $\sigma_\varepsilon$ | | | | | | |
| 0.1 | 0.003 | 0.003 | **1.10** | 0.178 | 0.183 | **1.03** |
| 0.2 | 0.003 | 0.004 | **1.34** | 0.175 | 0.192 | **1.09** |
| 0.3 | 0.003 | 0.005 | **1.62** | 0.174 | 0.208 | **1.19** |
| 0.4 | 0.003 | 0.006 | **1.99** | 0.174 | 0.228 | **1.31** |
| 0.5 | 0.003 | 0.007 | **2.37** | 0.175 | 0.252 | **1.44** |
| 0.6 | 0.003 | 0.008 | **2.79** | 0.172 | 0.276 | **1.60** |
| 0.7 | 0.003 | 0.009 | **3.32** | 0.177 | 0.305 | **1.72** |
| 0.8 | 0.003 | 0.010 | **3.83** | 0.165 | 0.330 | **2.00** |
| 0.9 | 0.002 | 0.011 | **4.46** | 0.177 | 0.366 | **2.07** |
| 1 | 0.002 | 0.012 | **5.12** | 0.171 | 0.391 | **2.28** |
| | | | **(C) Negative binomial** | | | |
| $k$ | | | | | | |
| 5 | 0.003 | 0.006 | **2.02** | 0.176 | 0.243 | **1.38** |
| 4.47 | 0.003 | 0.007 | **2.13** | 0.176 | 0.251 | **1.42** |
| 3.94 | 0.003 | 0.007 | **2.21** | 0.176 | 0.258 | **1.46** |
| 3.41 | 0.003 | 0.007 | **2.36** | 0.177 | 0.269 | **1.52** |
| 2.89 | 0.003 | 0.008 | **2.57** | 0.180 | 0.287 | **1.59** |
| 2.36 | 0.003 | 0.009 | **2.79** | 0.184 | 0.308 | **1.67** |
| 1.83 | 0.003 | 0.010 | **3.13** | 0.180 | 0.334 | **1.85** |
| 1.3 | 0.003 | 0.012 | **3.74** | 0.183 | 0.384 | **2.11** |
| 0.77 | 0.003 | 0.015 | **4.82** | 0.186 | 0.481 | **2.58** |
| 0.25 | 0.003 | 0.030 | **9.26** | 0.200 | 0.899 | **4.49** |

of overdispersion. However, the standard errors of these estimates were greatly reduced relative to the OLRE model (Table 1B). Modeling the most extreme case of zero inflation

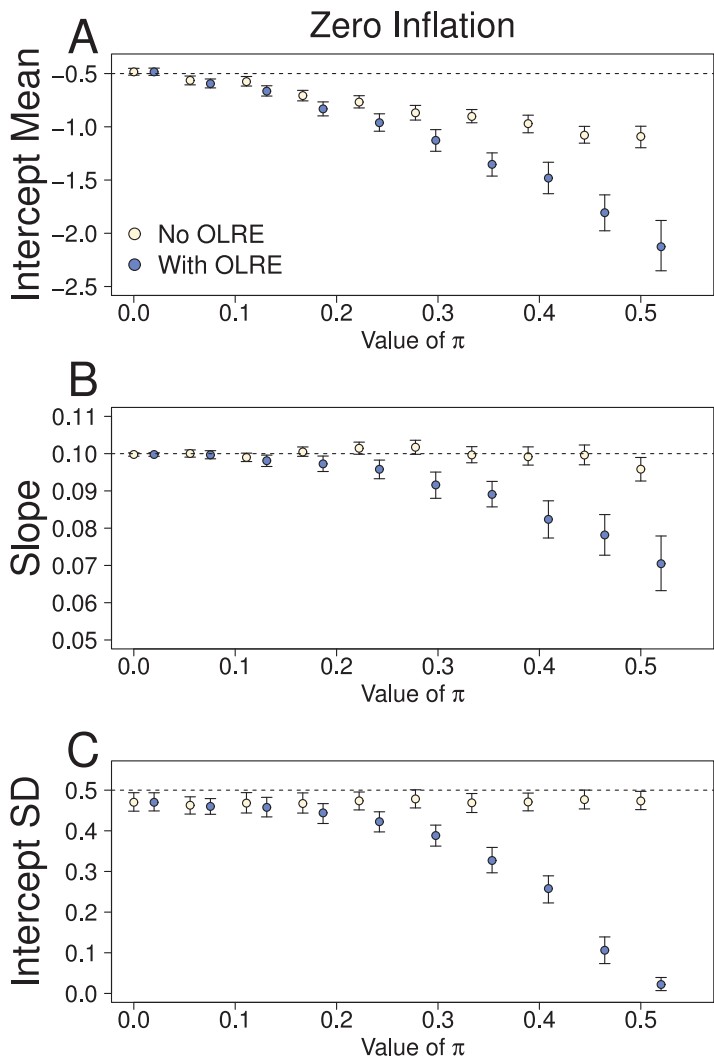

**Figure 2 Parameters for the intercept mean (A), slope of the effect of body size (B) and intercept standard deviation (C) generated under various levels of overdispersion for the Zero-Inflation simulations.** Light circles represent the mean values of the Naive models, where overdispersion was ignored. Blue circles represent the models containing observation-level random effects. Error bars are 95% confidence intervals of the mean as estimated by bootstrapping. Dashed horizontal lines denote the true (simulated) parameter values.

in the current dataset ($\pi = 0.5$) explicitly with zero-inflated Poisson (ZIP) models instead of using OLRE resulted in the recovery of both accurate parameter estimates ($\beta = 0.1$, SE = 0.005) and a correct estimate of the degree of zero inflation ($\pi = 0.49$, SE = 0.02).

## Negative binomial simulations

At moderate to high levels of overdispersion (dispersion parameter 5–20), the Naive and OLRE models produced similar parameter estimates (Fig. 3). It was only at extreme levels of overdispersion (>20) that estimates began to diverge, especially for $\mu_\alpha$, which became strongly negatively biased in the OLRE models (Fig. 3A), and $\sigma_\alpha$ which decreased rapidly

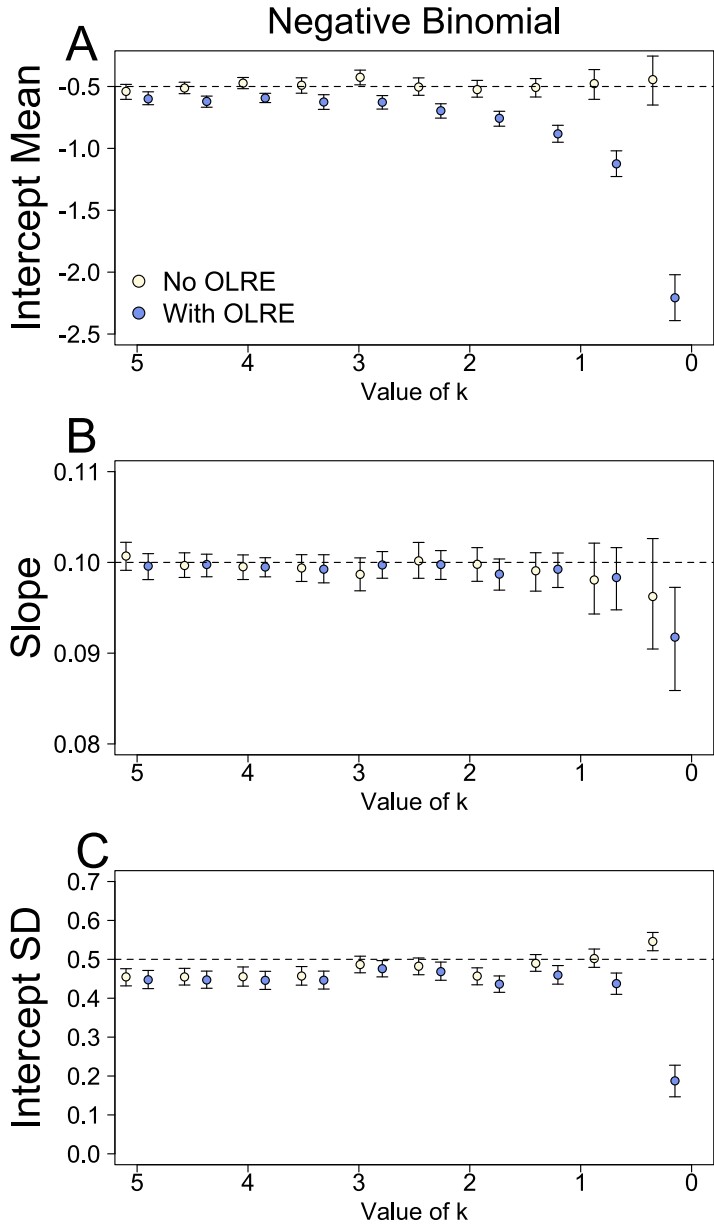

**Figure 3 Parameters for the intercept mean (A), slope of the effect of body size (B) and intercept standard deviation (C) generated under various levels of overdispersion for the negative binomial simulations.** Light circles represent the mean values of the Naive models, where overdispersion was ignored. Blue circles represent the models containing observation-level random effects. Error bars are 95% confidence intervals of the mean as estimated by bootstrapping. Dashed horizontal lines denote the true (simulated) parameter values.

towards zero (Fig. 3C). The Naive models recovered the correct parameter estimates at all but the most extreme case of overdispersion, but the standard errors of the estimates of all three parameters were greatly reduced in the Naive models by up to a factor of 9 (Table 1C).

## Overdispersion and $r^2$

There were stark differences between the Naive and OLRE models in the estimates of marginal $r^2$ (proportion of variance explained by the fixed effects). For all three sets of simulations, low levels of overdispersion yielded comparable estimates of $r^2$. As overdispersion increased however, the marginal $r^2$ of the Naive models remained essentially constant ($\sim$0.35), but decreases rapidly to zero in the OLRE models under all three scenarios (Figs. 4A–4C). Conversely models containing an OLRE had much higher values of conditional $r^2$ (proportion of variance explained by the fixed and random effects) than the Naive models containing no OLRE in all three simulations (data not shown).

Plotting the different components of the $r^2$ calculation separately revealed the reasons behind the divergent behaviour of the $r^2$ measures between the Naive and OLRE models. Figure 5 shows the behaviour of the estimates of fixed effect, random effect and residual variance at different levels of overdispersion for the Noise simulations. Data for the Zero-inflation and Negative Binomial simulations are similar and not shown here. For Naive and OLRE models, the fixed effects variance remained stable at all levels of overdispersion (Fig. 5A). Conversely, the random effects variance increased rapidly in tandem with overdispersion for OLRE models, but remained constant in the Naive models (Fig. 5B). There was a slight tendency for the residual variance to decrease in tandem with overdispersion in Naive models, but remained relatively constant for OLRE models (Fig. 5C).

## DISCUSSION

This paper used a simulation approach to test the ability of observation-level random effects (OLRE) to minimise overdispersion and thus recover unbiased parameter estimates in Poisson mixed effects models. Simulations revealed that the appropriateness of employing OLRE in mixed models depends on the process generating the overdispersion in the data: whilst OLRE perform well in minimising bias in cases where the data contain random extra-Poisson noise, they increased the degree of bias in zero-inflated data compared to models where the overdispersion was simply ignored. In addition, OLRE appeared to perform well when modelling aggregated count data at all but the highest levels of overdispersion. For all scenarios, ignoring overdispersion lead to greatly reduced standard errors of parameter estimates. Finally, a critical outcome of the simulations was to reveal the effect of overdispersion on measures of $r^2$. Irrespective of the cause of overdispersion, failing to account for overdispersion in the models resulted in greatly inflated estimates of the proportion of variance explained by the fixed effects, relative to cases where an OLRE was added to the models. Collectively, these results suggest a clear need for researchers to ascertain both the magnitude and source of overdispersion in biological datasets in order to derive accurate estimates of effect size and $r^2$.

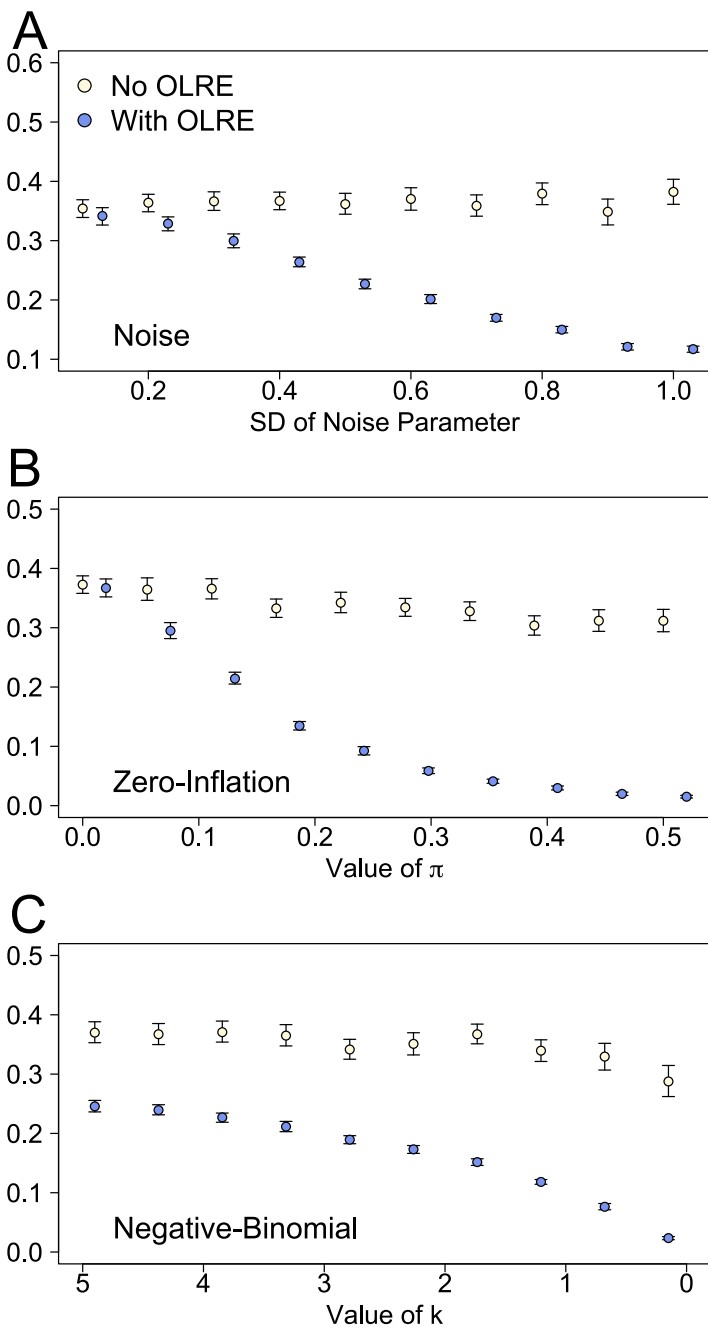

**Figure 4 Marginal $r^2$ values of models generated under 3 scenarios of overdispersion: (A) extra-Poisson noise; (B) zero-inflated data; and (C) data generated from a negative binomial distribution.** Moving from left to right on the $x$ axes corresponds to increasing levels of overdispersion in the models. Light circles represent the mean values of the Naive models, where overdispersion was ignored. Blue circles represent the models containing observation-level random effects. Error bars are 95% confidence intervals of the mean as estimated by bootstrapping. Ignoring overdispersion under all three scenarios resulted in greatly inflated estimates of the proportion of explained variance relative to where overdispersion was taken into account.

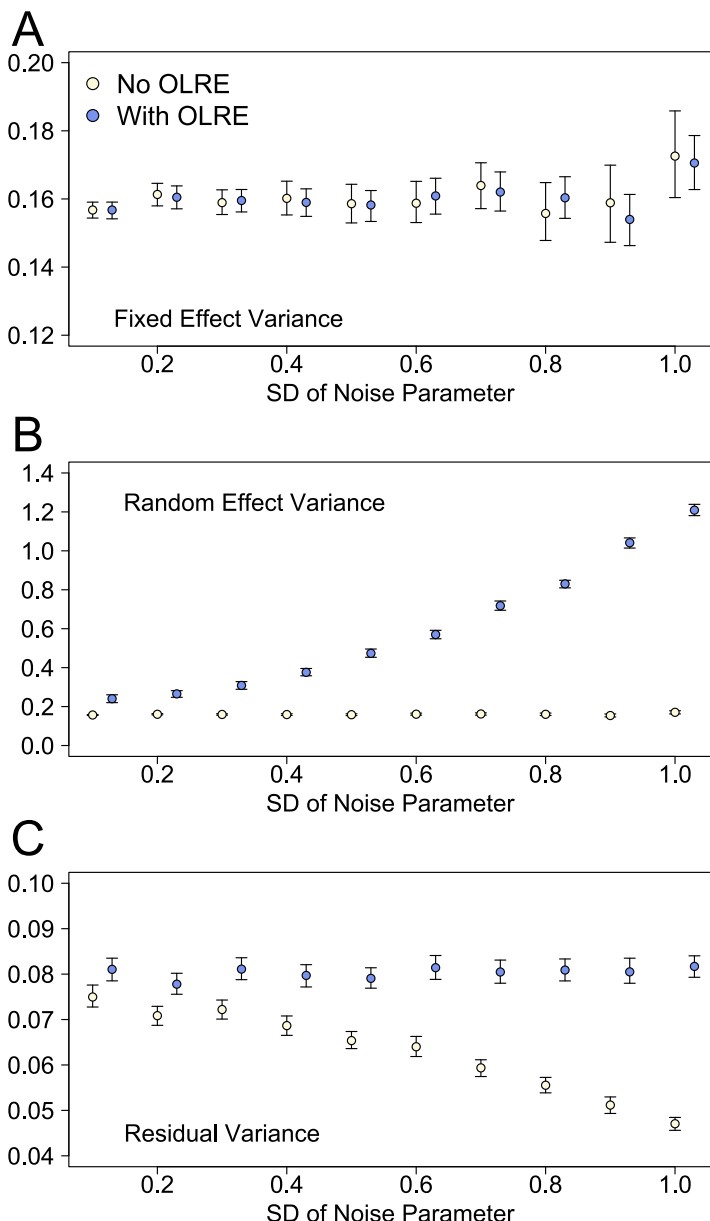

**Figure 5** **The three components of variance used to calculate the $r^2$ metrics proposed by** *Nakagawa & Schielzeth (2013)* **for the noise simulation datasets at various levels of overdispersion.** Light circles represent the mean values of the Naive models, where overdispersion was ignored. Blue circles represent the models containing observation-level random effects. Error bars are 95% confidence intervals of the mean as estimated by bootstrapping.

## Overdispersion and biased parameter estimates

Simulation results suggested that the ability of OLRE to minimise bias in parameter estimates in overdispersed count data depends on the process that generated the overdispersion. In cases where data contain extra-Poisson noise, adding OLRE to models results in the recovery of more accurate parameter estimates compared to when overdispersion is

simply ignored, a finding that is broadly congruent with similar investigations by others (*Kery, 2010*; *Zuur, Saveliev & Ieno, 2012*). It is perhaps reassuring, and not surprising, that modeling the data with the same structure that was used to generate the data recovers correct, unbiased estimates (see also *O'Hara & Kotze, 2010*). Similarly, employing OLRE to model aggregated data in the Negative Binomial simulations appeared to robustly estimate the correct parameters at all but the highest levels of overdispersion.

When the data are zero-inflated, OLRE fail to accurately estimate the model parameters at even moderate levels of overdispersion, and in some cases increased the bias relative to when overdispersion was ignored. Inspection of the model parameters indicated that as overdispersion increased, the magnitude of the SD of the OLRE random effect ($\sigma_\varepsilon$) increased rapidly, whilst the SD of the population random intercept term ($\sigma_\alpha$) decreased to zero. Therefore, it appears that even at moderate levels of overdispersion caused by zero-inflation, the model is unable to correctly partition variance between the $\sigma_\varepsilon$ and $\sigma_\alpha$, leading to inflated variance ratios ($\sigma_\varepsilon/\sigma_\alpha$) and biased parameter estimates. Conversely, when a model designed specifically to estimate the degree of zero-inflation was employed, the true parameter estimates were recovered even at the most extreme level of zero-inflation employed in the simulations ($\pi = 0.5$). Collectively these results suggest that OLRE are not an appropriate tool for modelling overdispersion when it is due to zero-inflation, and that researchers should instead employ modelling tools designed specifically to simultaneously model the effects of covariates of interest and quantify the degree of zero-inflation in the data e.g., zero-inflated Poisson (ZIP) and zero-inflated negative binomial (ZINB) models (see *Kery, 2010*; *Zuur et al., 2009*; *Zuur, Saveliev & Ieno, 2012*). Extra caution must be employed when adopting these approaches however, as one must be careful to distinguish between 'true' and 'false' zeroes in datasets, and adjust the modelling framework accordingly. *Martin et al. (2005)* demonstrated how the appropriate choice of model (ZIP, ZINB or Poisson) depends on the source of the excess zeros in the dataset, with poorly chosen models leading to impaired predictive performance of models, and inaccurate estimates of effect sizes.

In these simulations, applying the same model to data generated by either a Poisson or Negative Binomial process recovered similar parameter estimates. However, it is worth noting that Poisson and Negative binomial distributions make fundamentally different assumptions about the relationship between the mean ($\mu$) and variance ($v$): whilst $v = \mu$ in Poisson models, $v = \mu + \mu^2/k$ for Negative Binomial models (Eq. (12)), i.e., the variance increases non-linearly with the mean in the latter case (see *Ver Hoef & Boveng, 2007*; *O'Hara & Kotze, 2010*). Distinguishing between these two mean–variance relationships can be critical, as model mis-specification can lead to different parameter estimates in either case (*Ver Hoef & Boveng, 2007*), although this is not always true (*O'Hara & Kotze, 2010*, this study). *O'Hara & Kotze (2010)* point out that curvature in a plot of $(y_i - \mu_i)^2$ against $\mu_i$ from a Poisson model would potentially suggest a negative binomial model to be a better fit. If employing OLRE in a model and the suggested plot demonstrates curvature, it may be prudent to compare estimates of the OLRE Poisson model with a model using a negative binomial error structure to check that they do not differ considerably.

## Overdispersion and $r^2$

The most striking result from the modelling simulations is the effect of overdispersion on measures of explained variance. Simulations revealed that ignoring overdispersion leads to greatly inflated measures of the proportion of explained variance, as determined using a recent metric of $r^2$ proposed for mixed models (*Nakagawa & Schielzeth, 2013*). Conversely, when accounting for overdispersion using OLRE, marginal $r^2$ measures declined rapidly in tandem with overdispersion. This result is intuitive—as overdispersion increases, so too does the amount of unexplained variation in the data, and one would thus expect the $r^2$ to decrease assuming the model structure remains unchanged (i.e., not adding additional predictors). That marginal $r^2$ remains constant at all levels of overdispersion in the Naive models arises because the estimate of residual (unexplained) variance does not increase in line with overdispersion as perhaps it should. Conversely, the random effects variance component greatly increases in the OLRE models due to the increasing magnitude of $\sigma_\varepsilon$. Thus, when one calculates marginal $r^2$ as fixed effect variance/(fixed effect + random effect + residual variance), the denominator increases greatly in the OLRE models, whilst the numerator remains fairly stable, leading to a reduction in $r^2$ with increasing overdispersion.

With respect to conditional $r^2$, however, the OLRE models provide a cautionary note; because the random effects variance increases so much when adding an OLRE (as described above), calculating the variance explained by both the fixed and random effects [(fixed effect + random effect variance)/(fixed effect + random effect + residual variance)] leads to conditional $r^2$ values of near 0.9. This value is misleading, because the OLRE is basically a nuisance parameter that has little biological meaning (although variance ratios using OLRE are informative e.g., in calculating aggregation parameters, *Elston et al., 2001*), and yield little insight into the predictive power of the model. Collectively, these simulations suggest one should never report marginal $r^2$ values for Poisson mixed effects models without first having quantified and controlled for overdispersion, but also that one should avoid reporting conditional $r^2$ in models containing OLRE.

## Summary

This study adds to the body of literature demonstrating that ignoring overdispersion in mixed models can impair our ability to identify the true biological drivers underlying observed data. The principal novelty of this study is to demonstrate that ignoring overdispersion leads to inflated $r^2$ values, which may cause researchers to identify predictors as having a biologically meaningful effect when in fact they do not. Indeed, ignoring overdispersion during model selection can result in the retention of overly complex models and thus erroneous conclusions regarding which predictors significantly influence the outcome variable of interest (*Richards, 2008*). The utility of OLRE is their ability to account for overdispersion without making implicit, potentially erroneous, assumptions about the process that generated that overdispersion in the first instance (e.g., that the data arise from a negative-binomial distribution). However, these results suggest that OLRE cannot cope with overdispersion generated by zero-inflation, and that researchers should employ alternative modeling tactics for zero-inflated data (e.g., *Martin et al., 2005*). Moreover, all

three sets of simulations suggested OLRE should be used with caution when high levels of overdispersion are present (dispersion parameter >20). Collectively, these data highlight the necessity of quantifying and accounting for overdispersion in statistical models of count data in order to recover correct and unbiased estimates of model parameters and $r^2$ values. Ideally, model results and $r^2$ values reported in research papers should be accompanied by statistics regarding the degree of overdispersion present in candidate models in order to ensure that those results are both robust and interpretable.

## ACKNOWLEDGEMENTS

XH thanks Ben Bolker for advice regarding parametric bootstrapping in R. This manuscript benefitted greatly from several lengthy chats with Richard Inger about the vagaries of observation-level random effects, for which I am extremely grateful.

### Funding

This work was supported by a Research Fellowship awarded to XH by the Zoological Society of London and a British Ecological Society Research Grant awarded to XH. The funders had no role in study design, data collection and analysis, decision to publish, or preparation of the manuscript.

### Grant Disclosures

The following grant information was disclosed by the author:
Zoological Society of London.
British Ecological Society: 4720/5758.

### Competing Interests

Xavier A. Harrison is an employee of the Institute of Zoology, Zoological Society of London. XH declares no competing interests.

### Author Contributions

- Xavier A. Harrison conceived and designed the experiments, performed the experiments, analyzed the data, contributed reagents/materials/analysis tools, wrote the paper, prepared figures and/or tables, reviewed drafts of the paper.

### Data Deposition

The following information was supplied regarding the deposition of related data:

All R code for the simulations and graphs in this manuscript are available online at FigShare. DOI:

http://dx.doi.org/10.6084/m9.figshare.1144471

For a basic guide of how to quantify overdispersion in mixed models, and fit observation level random effects, see the 'Step by Step Guide' R code and accompanying 'Overdispersion Data File'. Individual R scripts for the Noise, Zero-Inflation and Negative Binomial Simulations are also provided.

## Supplemental Information

Supplemental information for this article can be found online at http://dx.doi.org/ 10.7717/peerj.616#supplemental-information.

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
