# Peer review of "Using observation-level random effects to model overdispersion in count data in ecology and evolution"

_PeerJ, doi:10.7717/peerj.616_

## Round 0.1 · original submission · Major Revisions

Two Reviewers have provided their comments about this manuscript. If you wish to revise your manuscript, please take the referee comments fully into account and provide point-by-point responses with a full list of changes. Revisions in this category may involve substantial text changes, recalculations or new analyses in addition to more minor clarifications and corrections. Other comments of the reviewers are relevant as well and you should follow them.

·

Basic reporting

The researcher aims to addresses the problem of overdispersion. The manuscript pointed out that there is a positive relationship between the magnitude of overdispersion and bias in the parameter estimation. In the conclusion, it shows the use of observation-levels random effect can compensate the overdispersion in count data. It also points out weakness of the method’s ability.

I can observe the research topic fall right in the scope of PeerJ. The conclusions are interesting and will be beneficial to its readership. However, properly used technical terms and mathematical notations will surely elevate the contribution. While this manuscript is mainly contributed to statistical community and other research communities that encounter the problems, I think it is my obligation to point out several misused statistical notations.

Experimental design

1. While in the manuscript this method is only tested on simulated datasets, and provided no mathematical derivation, I think a case study on real-observation biological data will elevate the value of the paper.
2. Line 200-214: The “OLRE” and “Naïve” GLMM models may need more explanation, given it will be used throughout the rest of the manuscript.
3. Line 217-219: Since the marginal r2 and conditional r2 will be heavily discussed in the following context, please give a equation for calculating the value.

Validity of the findings

R result are reproducible.

Additional comments

Overall, I hope to see properly-formulated equations for the values in the figures and tables. I suggest the author improve this article by presenting the idea in a mathematical manner.

Section and subsection did not numbered?

Line 75, 77, 98: degree of freedom should be abbreviated d.f.

Line 106, Generalized linear mixed models

Line 140~143: please use subscript C_ij, also , Normal distribution N(..,..) , also, please add eqn. number for each equation.

Line 151: hard to follow what is in the parenthesis.

Line 161, 162, Poison distribution should be: Pois(...)

Reviewer 2 ·

Basic reporting

At lots of places active voice has been used, in my personal opinion, the use of active voice should be avoided in the research documents. If possible, avoid active voice.

Abstract:
Kindly mention in the abstract that OLRE represents Observation-Level Random Effects.

Last paragraph of the abstract is looking like abstract of abstract. The sentences in this paragraph are useful, but they should be used at appropriate place in the abstract.

Introduction:
Line, "Perhaps the most common method employed to model count data is to assume the data approximate a Poisson distribution, and specify statistical models accordingly" require a citation.

A brief description of other tools other than OLRE in the introduction section could help the reader.

Expand GLM in line 72.

Explain in line 82, why the sum of squared Pearson residuals should approximate a Chi-squared distribution ..., or provide some reference.

Expan SS in line 95.

Maybe passive voice would be better in line 114.

Period (.) is missing in line 131.

In line 172, the statement, "This is the most straightforward zero-inflated data one can generate, ..." is very arrogant, unless supported by some facts or reference. This sentence should be re-phrase.

Experimental design

The experimental design is good, but a lack of clarity in the 'Methods' section is creating confusion. A table explaining all the scenarios and the values of the parameters used for them would help reader better understand the differences among them. It is also not mentioned why certain values of the parameters are choosen e.g. in line 170 it is mentioned that, "I simulated 10 scenarios where ψ ranged from 0.5 to
1 in increments of 0.05" but it is not explained why only starting from 0.5 to 1, why not lower than 0.5 or higher than 1?

Validity of the findings

The validity of the findings would depend upon the comments made in the "Experimental Design" section. Unless it is clearly mentioned why certain range of the parameter is choose, the findings can not be assesed.

Additional comments

The attempt to model the overdispersion using OLRE would be quiet useful while dealing with the overdispersed data in estimation of parameters. The presented experiment is also good, but I feel that there is a need to clarify the methodolgy used, and support the use of parameters used either with reference or with some reasoning.

I am recommending the major revisions as I could not understand the methodology section clearly. So, I avoided making any comments on the findings. If you make this section more clear, I feel after that the article should be accepted.

---

## Round 0.2 · accepted · Accept

I read carefully this updated version of the manuscript and the related rebuttal. Authors kindly addressed all the reviewers' points and, in particular, made a much better job in explaining the methodology. A significant improvement also comes from a better presentation of previous works, methodologies, and discussions provided in other papers. I am pleased to say that the revised version of the manuscript is suitable for publication.